# Overview of the Efficacy of Using Probiotics for Neurosurgical and Potential Neurosurgical Patients

**DOI:** 10.3390/microorganisms12071361

**Published:** 2024-07-02

**Authors:** Sabina Fijan, Tomaž Šmigoc

**Affiliations:** 1Faculty of Health Sciences, University of Maribor, Žitna ulica 15, 2000 Maribor, Slovenia; 2Department of Neurosurgery, University Medical Centre Maribor, Ljubljanska ulica 5, 2000 Maribor, Slovenia; tomaz.smigoc@ukc-mb.si

**Keywords:** neurosurgery, probiotics, synbiotics, microbiota, brain injury, neurologic injury

## Abstract

This review delves into the emerging field of the gut microbiota–brain axis, emphasizing its bidirectional communication and implications for neurological health, particularly in trauma and neurosurgery. While disruptions in this axis can lead to dysbiosis and hinder neurological recovery, recent studies have highlighted the therapeutic potential of interventions like probiotics in targeting this axis. This review aims to focus on the efficacy of probiotic supplementation to support the gut microbiota–brain axis in trauma, neurosurgery, or pain based on the current clinical trials to assess the complex interplays among probiotics, the gut microbiota, and the central nervous system (CNS). This comprehensive literature review identified 10 relevant publications on probiotic interventions for various neurosurgical conditions across multiple countries. These studies demonstrated diverse outcomes, with significant improvements observed in gastrointestinal mobility, inflammatory responses, and infection rates, particularly in post-traumatic brain injury and spinal surgery. Probiotics also showed promise in mitigating antibiotic-associated diarrhea and modulating inflammatory cytokines. Despite the promising findings, the complex interplays among probiotics, the gut microbiota, and the central nervous system (CNS) call for cautious interpretation. Conflicting outcomes emphasize the need for better-designed trials to understand strain-specific and disease-specific effects accurately. In conclusion, probiotics offer a promising adjuvant therapy for neurosurgical patients, traumatic brain injuries, and post-spinal surgery. However, further well-designed randomized controlled trials are essential to elucidate the intricate relationship between microbiome-modulating interventions and the CNS via the gut microbiota–brain axis.

## 1. Introduction

Neurological surgery was recognized as a specialty at the beginning of the 20th century. The early neurosurgeon would have been a clinician–neurologist who would diagnose and localize brain lesions based on clinical history and neurological examination. The clinical–pathological correlation of brain lesions has led to the diagnosis of brain tumors, vascular lesions, and spinal disorders [1]. The development of microscopes, micro-instruments, diagnostic modalities (like computer tomography (CT), magnetic resonance imaging (MRI), new technology in the fields of neurostimulation, genetics, and microbiology have since advanced neurosurgery. Neurosurgery primarily focuses on the surgical treatment of the brain and spine. The brain, which generates and controls all behavior, is composed of billions of neuronal and non-neuronal cells interconnected by complex structural networks [2]. The spine is composed of vertebra, cartilage tissue, ligaments, muscles, and the spinal cord. The functions of the spine include protecting the spinal cord; supporting the body’s weight; enabling trunk motion through its flexibility; and providing robust attachment points for trunk and limb muscles [3].

Traumatic brain injury is one of the most serious types of trauma that imposes heavy social and economic burden on healthcare systems worldwide [4]. Degenerative and inflammatory spine diseases affect nearly 90% of people at some point in their lives and are the leading cause of chronic low back pain [5]. Spine degeneration impacts the mechanical properties and anatomic morphology of the vertebrae, discs, and surrounding soft tissues [3]. Motion segment stiffness first decreases, then increases after more advanced stages of degeneration [6]. These changes can alter the range of motion, loading patterns, and tolerance to traumatic events involving loads significantly greater than those encountered during common activities [3].

Although the connection between probiotics and neurosurgery is not obvious at first glance, there is already ancient wisdom of gut microbiota and brain connections that have been established as an important bidirectional communication, which plays a critical role in regulating not only gastrointestinal homeostasis but has also been linked to higher emotional and cognitive functions [7]. The gut microbiome interacting with most human cells is now considered an important ally due to increasing knowledge about the important and essential role of the gut microbiome in human health [8,9]. The recent discovery of interconnections within the human body has unveiled numerous research avenues, including the ‘gut microbiota–brain axis’, ‘gut microbiota–skin axis’, ‘gut microbiota–liver axis’, ‘gut microbiota–sexual hormones axis’ and ‘gut microbiota–lung axis’. These findings highlight crucial links between the gut microbiota and the immune system, the central nervous system (CNS), and the endocrine system, playing pivotal roles in sustaining systemic homeostasis [9,10,11,12,13]. Different pathways of the gut microbiota include the trimethylamine (TMA)/trimethylamine N-oxide (TMAO) pathway, short chain fatty acids (SCFAs) production, and secondary bile acid pathways [2,14].

The gut microbiota–brain axis therefore involves bidirectional communications, metabolite production, and modulation of immune and neuronal functions. The microbiome plays two important and distinct roles. On the one hand, it beneficially modulates the immune system, the enteric nervous system, sympathetic and parasympathetic pathways of the autonomic nervous system, and neuronal functions using the neural, endocrine, immune, and metabolic pathways. Molecules, produced by the gut microbiota, such as tryptophan metabolites, vitamins, and SCFAs that regulate G-protein-coupled receptors to mediate neurotransmitter release (serotonin, dopamine, noradrenaline, γ-aminobutyric acid, acetylcholine, and histamine), contribute to neuronal and glial cell function. Some of the smaller molecules can even reach the brain despite the highly selective blood–brain barrier. Gut–brain communication also occurs via the vagus nerve [2,10,15,16]. However, on the other hand, after a central nervous system injury, such as brain trauma or brain surgery, several inflammatory changes, including the necrosis and apoptosis of neuronal tissue, propagate downward inflammatory signals to disrupt the microbiome homeostasis. Damage to the central nervous system can impair the gastrointestinal function through the hypothalamic–pituitary–adrenal axis and the vagus nerve, causing intestinal inflammation. In turn, the dysfunction of the gastrointestinal tract can worsen brain damage. Microbiome dysbiosis can therefore impact the upward signaling to the brain and interfere with recovery in the neuronal functions and brain health. [7,15,17,18]. This bidirectional communication via the spinal cord–gut immune axis also influences spinal cord injuries as the damage extends past paralysis and includes other debilitating outcomes such as immune dysfunction and gut dysbiosis due to the disruption in the intestinal barrier, thereafter triggering neurogenic inflammatory responses that impede recovery [19,20,21,22]. Fecal microbiota transplantation in a rat model study regulated the gut microbiota–spinal cord axis and improved spinal pathology, reducing post-injury inflammation to promote recovery after a spinal cord injury [23]. 

A dysbiotic gut could possibly even be connected to lower back pain or osteoarthritis as various disease pathogenesis have recently been revisited with plausible cross-talks and micromolecular mimicry. For example, although traditional teachings have noted that the central nervous system and intervertebral disc (IVD) are sterile, recent 16S rRNA molecular methods have found the presence of a microbiome in these structures [24]. Normal IVD microbiome is connected to a normal gut microbiome. It was found that normal IVD contained, in abundance, the phyla Bacillota and Actinomycetota (previously, Firmicutes and Actinobacteria) [25] that are SCFA producers, whilst the translocation of pathogens from the gut and skin was associated with infection, inflammation, and disc degeneration [24]. Osteoarthritis pain is intricately associated with the activation of the innate immune system and subsequent inflammatory response involving changes in the gastrointestinal microbiome and thus causing a disruption of gut permeability [26]. 

Elucidating the interconnected pathways among the microbiota, spinal cord, brain, and immune and nervous systems may therefore facilitate novel treatment strategies [23]. The possible therapeutics that could reduce the adverse contributions of the gut microbiota to the late sequelae of traumatic brain injury and spinal cord injury include anti-inflammatories, antibacterials, antivirals, and probiotics [27,28]. Therefore, effective new technologies and preventive strategies for feeding intolerance among patients with severe traumatic brain injury have great potential to be applied in practice [29]. 

Probiotics, defined as ‘live microorganisms that confer a health benefit, when administered in adequate amounts’, are therefore one of the promising therapeutic approaches to maintaining neurological health [30]. Prebiotics, as substrates selectively utilized by host microorganisms thus conferring a health benefit, and synbiotics, that combine probiotics and prebiotics with synergistic mechanisms of action [31,32], are also promising options for maintaining neurological health. Probiotics have been used extensively in a variety of medical conditions, and numerous well-designed, randomized, controlled trials have proven that they improve symptoms of diarrhea, gastroenteritis, irritable bowel syndrome, inflammatory bowel disease, cancer, depressed immune function, inadequate lactase digestion, infant allergies, failure-to-thrive, hyperlipidemia, hepatic diseases, *Helicobacter pylori* infections, genitourinary tract infections, as well as many others [33,34,35,36]. Probiotics have also been proven to support the treatment of critically ill patients or patients after multiple traumas, including neurosurgical patients, with acute neurohormonal and inflammatory cascades and immediate immune responses [37,38,39]. This review aims to focus on the efficacy of probiotic supplementation to support the gut microbiota–brain axis in trauma, neurosurgery, or pain based on the current clinical trials to assess the complex interplays among probiotics, the gut microbiota, and the central nervous system (CNS). 

## 2. Materials and Methods

A literature overview was conducted to summarize the existing studies of using probiotics for neurosurgical patients. We used the search strategy noted in Table 1. The results are presented in a tabular and descriptive format. For each publication, data about the author(s), publication year, aim, probiotics used, and main findings were extracted. 

## 3. Results

Using the inclusion criteria, there was a total of 10 studies utilizing probiotics for neurosurgical conditions [17,40,41,42,43,44,45,46,47,48]. The main findings of the 10 included studies are noted in Table 2, Table 3 and Table 4.

The findings of the 10 studies noted in Table 2, Table 3 and Table 4 were very diverse after neurosurgery, traumatic brain injury, spinal surgery, or back pain. Of these, two studies involved the utilization of probiotics after neurosurgery [17,40], four were conducted after traumatic brain injury [41,42,43,44], and four after spinal surgery or back pain [45,46,47,48]. The studies were conducted in Australia, Brazil, China, Denmark, India, Japan, and Greece. Two studies by Tan et al. were found [43,50]; however, it seems that both are part of the same clinical trial. The former publication is in Chinese; therefore, it was omitted. 

On the one hand, some of the studies found the following statistically significant differences in the probiotic group compared to the control group: improved gastrointestinal mobility after neurosurgery [17], reduced inflammatory response or increased proinflammatory response of patients with TBI and back pain [41,42,43,48], duration in hospital or ICU of patients with brain injury [42,44], reduced infection rates of patients with TBI [42,44], GCS scores of patients with TBI [42], decreased presence of pathogenic microbes in gut microbiome of patients after spinal surgery [45], and lower back pain after one-year follow-up [47]. A lower, but not statistically significant, incidence of neurosurgical site infections [40,43], rate of complications of patients with TBI [41], ICU duration of patients with TBI [43], and pain score of patients with back pain [46,47] were found in the probiotic group compared to the control group. 

On the other hand, other studies showed that no significant differences were found for diarrhea, nausea or vomiting of patients after craniotomy for brain tumors [17], and six-point composite disease activity index (mJSpADA) of children with pain and back pain between both groups [48].

Eight studies were double-blind, randomized, controlled clinical trials [17,40,41,42,43,45,47,48], one study was a controlled clinical trial without blinding information [44], and one study was a case study [46]. A total of 642 patients are included in Table 1, Table 2 and Table 3, respectively, where 316 are in the probiotic group and 326 in the control group.

Three studies utilized single-strain probiotics including the following: *Lactobacillus johnsonii* La 1 [44], *Lacticaseibacillus rhamnosus* GG [47], and *Enterococcus faecium* 129 BIO 3B-R [45]. The remaining seven studies involved multi-strain probiotics that contained various strains of *Bifidobacterium*, *Lactobacillus*, *Lacticaseibacillus*, *Lactiplantibacillus*, *Limosilactobacillus*, *Streptococcus*, *Enterococcus*, and *Saccharomyces* genera [17,40,41,42,43,46,48]. Four studies that involved multi-strain probiotics did not report strain information [41,42,43,48]. 

## 4. Discussion

This overview on the effectiveness of probiotics in supporting the gut microbiota–brain axis in trauma, neurosurgery, or pain based on the current clinical trials has shown that probiotics can be effective in alleviating the various symptoms accompanying trauma or changes to the central nervous system. Although most clinical trials did not investigate the exact mechanisms of action of the probiotics, it was found that various single-strain and multi-strain probiotics seemed effective. 

Several systematic reviews have shown that probiotic supplementation can significantly improve gastrointestinal function and relieve functional constipation thus improving disease burden [51,52,53]. Gastrointestinal dysfunction is an important clinical problem in patients after any surgery; up to 80% of neurosurgical patients exhibit gastrointestinal dysfunction due to damage to the central nervous system that can impair the gastrointestinal function via the hypothalamic–pituitary–adrenal axis and the vagus nerve or the bidirectional gut microbiota–brain axis [17,18,54]. 

### 4.1. Probiotics and Gastrointestinal Health after Neurosurgery

The study by Jiang et al. investigated the influence of a multi-strain probiotic that contained six bifidobacteria strains (*Bifidobacterium animalis* subsp. *lactis* HNO19, *Bifidobacterium animalis* subsp. *lactis* BB-12, *Bifidobacterium. animalis* subsp. *lactis* Bi07, *Bifidobacterium animalis* B94, *Bifidobacterium bifidum* Bb06, *Bifidobacterium longum* R175), four *Lacticaseibacillus* strains (*Lacticaseibacillus rhamnosus* GG, *Lacticaseibacillus rhamnosus* R11, *Lacticaseibacillus*
*casei* Lc11, *Lacticaseibacillus paracasei* Lpc37), and five other strains (*Lactobacillus helveticus* R52, *Lactiplantibacillus plantarum* R1012, *Limosilactobacillus reuteri* HA188, *Lactobacillus acidophilus* NCFM, *Streptococcus thermophilus* St21) on gastrointestinal function in patients after craniotomy for brain tumors and found a significantly shorter time of the first stool and flatus [17]. In five studies that investigated the effects of probiotics on gastrointestinal health after a traumatic brain injury [41,42,43,44,50], the patients were in intensive care with enteral nutrition and usually lying for extended periods. However, reduced inflammatory response and enhanced immune function were observed in the probiotic groups compared to the placebo groups. The incidence rate of complications was also evidently lower in probiotic groups compared to the placebo groups. Thus, all five studies concluded that early enteral nutrition, combined with daily prophylactic administration of probiotics, was effective for patients with traumatic brain injury. This is in line with other clinical studies that have found that probiotics were effective in alleviating various postoperative complications related to gut health after various surgeries or procedures such as appendectomy, colorectal cancer surgery, ileostomy closure, and others [55,56,57,58,59].

### 4.2. Probiotics and Antibiotic-Associated Diarrhea after Neurosurgery

Antibiotic-associated diarrhea is also common in patients after neurosurgery [60]. Surgical antibiotic prophylaxis (SAP) is standard treatment recommended by the Centers for Disease Control [61] to prevent surgical site infections that are among the most common healthcare-associated infections. Although SAP is very important in preventing surgical site infections, there are several possible adverse effects of SAP including allergy, anaphylaxis, nausea, antibiotic-associated diarrhea, *Clostridioides difficile*-associated diarrhea, and antibiotic resistance [45,62]. Probiotics are a well-known adjuvant to antibiotic therapy that can prevent and alleviate antibiotic-associated diarrhea and *Clostridioides difficile* infection [63,64]. Strains of *Bacillus licheniformis*, *Bifidobacterium longum*, and *Bacillus subtilis* have demonstrated the greatest effect sizes in antibiotic-associated diarrhea compared with other probiotics. Other strains, including strains of *Lactobacillus acidophilus*, *Lacticaseibacillus casei*, and *Saccharomyces boulardii*, have demonstrated a moderate effect [64]. 

### 4.3. Probiotics and Surgical Site Infections after Neurosurgery

Probiotics reduce the risk of surgical site infections (SSI) and promote wound healing [65,66]. For example, perioperative supplementation with probiotics decreased postoperative infectious complications of colorectal surgery [9,67,68]. In the study by Kaku et al. [45], the probiotic strain *Enterococcus faecium* 129 BIO 3B-R was used to assess the influence on antibiotic-associated diarrhea and the reduction in SSI after spinal surgery. While they did not find significant differences in the alpha and beta diversity in the gut microbiome, it was found that *Streptococcus gallolyticus* and *Roseburia* spp. were significantly decreased in the probiotic group compared to the control group. *Streptococcus gallolyticus* is adversely associated with infections, sepsis, and even colorectal cancer [69,70,71]; therefore, probiotics can prevent or alleviate the possible adverse effects after spinal surgery. However, on the other hand, the modulation of gut microbiota is very complex and clinical studies should be conducted carefully and critically. Although the pathogenicity and function of *Roseburia* spp. has not been confirmed [45], the reduction in *Roseburia* spp. among other genera is associated with possible adverse effects including chronic pain, hypertension, and depressive disorders [72,73,74]. This indicates that careful probiotic selection is crucial and that the long-term effects of probiotics need to be investigated for the prevention of antibiotic-associated diarrhea after neurosurgery. Tzikos et al. [40] assessed the incidence of SSI after multi-trauma surgeries, including neurosurgeries, and found a statistically significant lower incidence of SSI in the probiotic group (*Lactobacillus acidophilus* LA-5, *Lactiplantibacillus plantarum* UBLP-40, *Bifidobacterium animalis* subsp. *lactis* BB-12, *Saccharomyces boulardii* Unique-28) compared to the placebo group. In the clinical studies by Wan et al., Tan et al., and Falcao de Arruda et al. [42,44,50], nosocomial infections were also lower in the probiotic groups compared to the placebo group of patients with brain injuries. Two of these three clinical trials utilized multi-species probiotics with no strain information—undefined strains of *Bifidobacterium longum*, *Lactobacillus bulgaricus*, and *Enterococcus faecalis* were used in the clinical trial by Wan et al. [42] and undefined strains of *Bifidobacterium longum*, *Lactobacillus bulgaricus*, and *Streptococcus thermophilus* were used in the clinical trial by Tan et al. [50]. The third clinical trial, conducted by Falcao de Arruda et al. employed the single-strain probiotic, *Lactobacillus johnsonii* La 1 [44]. 

### 4.4. Probiotics, the Inflammatory Response, and the Central Nervous System

It is well known that probiotics have a strong influence on the immune function through the gut-associated lymphoid tissue, by reducing inflammatory parameters and increasing proinflammatory parameters [13,75,76]. Three studies of our review assessed the influence of multi-strain probiotics on the inflammatory response in patients after traumatic brain injury and utilized multi-strain probiotics with undefined strains of *Bifidobacterium longum*, *Lactobacillus acidophilus*, *Lactobacillus bulgaricus*, *Enterococcus faecalis*, or *Streptococcus thermophilus* [41,42,43]. Zhang et al. [41] found a significant reduction in an inflammatory response (IL-6, IL-8, TNF-α), while Wang et al. [42] found a significant reduction in IL-6, IL-10, TNF-α, and CRP, and Tan et al. [43] found a significant reduction in IL-4, IL-10, as well as a significant increase in anti-inflammatory parameters, INF-γ and IL-12p70. Shukla et al. [48] investigated the influence of VSL#3 on the immune response in children with juvenile idiopathic arthritis (enthesitis-related arthritis), including back pain. Although they did not find significant differences in the improvement in the six-point composite disease activity index of modified juvenile spondyloarthropathy disease activity, a significant decrease in the inflammatory cytokine IL-6 was found. Brain and spinal cord injuries can trigger immunological dysfunctions, prompting neuroinflammation and the aberrant expression of inflammatory factors like chemokines, cytokines (e.g., IL-1β, IL-6, IL-10), and tumor necrosis factor (TNF-α). This cascade often involves peripheral cell recruitment and local tissue damage. By modulating the gut microbiota, it is possible to potentially manage neurological disorders linked with brain and spinal cord injuries and back pain, thereby addressing associated gut dysbiosis. The possible therapeutics that could reduce the adverse contributions of the gut microbiota to the late sequelae of traumatic brain or spinal cord injury include anti-inflammatories, antibacterials, antivirals, and probiotics. [7,15,28,42,77,78,79,80]. The results of the studies therefore confirm the positive influence of probiotic supplementation on reducing inflammatory response. Probiotics as well as synbiotics have also proven to benefit the recovery of the early inflammatory response and other immunomodulatory effects after various surgeries [57,81,82]. 

### 4.5. Other Indirect Influences of Probiotics on the Central Nervous System

Other important positive effects of probiotic supplementation of patients in our review included a shorter stay in the intensive care unit [43,50] [44] and reduced back pain [46,47]. Other studies have also found that probiotics could be associated with reducing pain after surgery [83], preventing and treating immune-related adverse events in novel immunotherapies against malignant glioma [75], reducing number of seizures [84], and reducing stress response and plasma C-reactive protein CRP in veteran patients with posttraumatic stress disorder and mild traumatic brain injury [85]. Many probiotics produce short chain fatty acids, which are speculated to play a key role in neuroendocrine-immune regulation. Although the underlying mechanisms through which SCFAs might influence brain physiology and behavior have not been fully elucidated [86], brain tumor patients lack SCFA-producing probiotics [87]. 

While the 2014 review by Curtis and Epstein [88] initially suggested that probiotic supplementation correlated with lower rates of infection in traumatic brain injury (TBI) and other trauma patients, the 2017 review by Brenner et al. [89] took a more cautious stance. Despite acknowledging some potential in probiotic and prebiotic interventions, Brenner and colleagues cautioned against overstating their benefits, particularly considering the disparity between popular press coverage and empirical evidence. In a more recent review by Noshadi et al., 2022 [90], the impact of probiotics on inflammatory parameters and length of stay in intensive care unit (ICU) among trauma and traumatic brain injury patients was researched. Notably, this review also included clinical trials utilizing synbiotics [39,91,92]. This inclusion, however, raises questions about whether the observed effects can be solely attributed to probiotics or also involve prebiotics [31,32]. While certain clinical trials within Noshadi’s review did show significant effects of probiotic or synbiotic supplementation on parameters such as CRP levels, interleukin Il-6, and ICU stay duration [91,92], a meta-analysis across all seven studies failed to identify significant effects of probiotics supplementation on these measures. Consequently, the authors cautioned that the limited number of randomized controlled trials (RCTs) and the overall sample size were insufficient to draw definitive conclusions. It is crucial to highlight that systematic reviews and meta-analyses frequently yield conflicting outcomes when evaluating probiotic efficacy. This discrepancy partly arises from a limited comprehension of the distinct characteristics of probiotic trials. Consequently, clinical decisions regarding probiotic use have been convoluted. Many trials fail to disclose strain composition, leading to varying conclusions regarding probiotic efficacy within a particular disease category. It is important to recognize that probiotic efficacy is not only strain-specific but also disease-specific, further complicating the interpretation of results [93].

### 4.6. Conclusions and Perspectives

Considering all this, probiotics could be an important adjuvant therapy for neurosurgical patients, patients with traumatic brain injuries, and after spinal surgery. However, more carefully designed randomized controlled trials are needed to further investigate the effect of probiotics in greater detail. The current studies have high methodological heterogeneity and small sample sizes, highlighting the need for well-designed and controlled studies to elucidate the complex linkage among the microbiome, microbiome-modulating interventions, and the central nervous system via the gut microbiota–brain axis.

Looking ahead, it is crucial to explore other axes within the human body such as the ‘gut microbiota–skin axis’, ‘gut microbiota–liver axis’, ‘gut microbiota–hormones axis’, ‘gut microbiota–lung axis’, and other microbiota–organs axes [94] to fully understand the role of probiotics in overall health. The firm connection between probiotics and these various axes underscores their potential in boosting health across multiple systems. Future research should continue to investigate these connections, aiming to develop targeted probiotic therapies that can enhance health and recovery in diverse medical contexts.

## Figures and Tables

**Table 1 microorganisms-12-01361-t001:** Research strategy containing inclusion criteria and search limits.

Databases:	PubMed, ScienceDirect and manual search
Search strategy:	“neurosurgery” OR “brain tumor” OR “brain tumor” OR “brain injury” OR “brain disease” OR “back surgery” OR “back surgery” OR “spinal surgery” OR “back pain” OR “neurosurgical” OR “neurologic injury”) AND (“probiotics”
Types of research:	Randomized, placebo controlled clinical trials, case studies
Language:	Publications in English
Exclusion criteria:	Studies that utilized synbiotics or prebiotics were excluded to isolate only the influence of probiotics. Publications in languages other than English
Timeframe:	Up to the 29th of February 2024

**Table 2 microorganisms-12-01361-t002:** Findings of the two studies investigating the effect of probiotics on patients after brain surgery, in descending chronological order.

Reference	Aim and Study Type	Participants Who Completed the Study	Intervention		Main Findings
			Probiotics	Dosage/Duration	
Jiang et al., 2023 [17], China	Double-blind RCT to assess the effect of probiotics on postoperative GI function in patients after craniotomy for brain tumors.	180 patients after craniotomy for brain tumors. 88 in probiotic group. 92 in control group.	*Bifidobacterium animalis* subsp. *lactis* HNO19, *Bifidobacterium animalis* subsp. *lactis* BB-12, *Bifidobacterium animalis* subsp. *lactis* Bi07, *Bifidobacterium animalis* B94, *Bifidobacterium bifidum* Bb06, *Bifidobacterium longum* R175, *Lacticaseibacillus * rhamnosus* GG, *Lacticaseibacillus* * *rhamnosus* R11, *Lacticaseibacillus* * *casei* Lc11, *Lactobacillus helveticus* R52, *Lacticaseibacillus* * *paracasei* Lpc37, *Lactiplantibacillus* * *plantarum* R1012, *Limosilactobacillus* * *reuteri* HA188, *Lactobacillus acidophilus* NCFM, *Streptococcus thermophilus* St21. (Zhongke Yikang Biological Technology, Beijing, China)	Dose: 4 g bid. Total cfu per day: 1.1 × 10^9^ cfu. Duration: 15 days.	The time of first stool and flatus were significantly shorter in the probiotics group compared to the placebo group (*p* < 0.001), suggesting that probiotics can improve the gastrointestinal mobility of patients who received craniotomy. No significant trends were observed for any other of the secondary outcome variables (assessments of the time of spontaneous bowel movements, diarrhea, nausea, and vomiting, changes in gastrointestinal permeability, and clinical outcomes).
Tzikos et al., 2022 [40], Greece	Double-blind RCT to assess the effect of probiotics against SSI of patients after MT surgeries, including neurosurgeries.	A total of 103 patients, of these, 19 patients after neurosurgery. 6 in probiotic group. 13 in control group.	*Lactobacillus acidophilus* LA-5, *Lactiplantibacillus* * *plantarum* UBLP-40, *Bifidobacterium animalis* subsp. *lactis* BB-12, *Saccharomyces boulardii* Unique-28. (LactoLevure^®^, Athens, Greece).	Dose: 2 sachets bid. Total cfu per day: 1.5 × 10^10^ cfu. Duration: 15 days.	This study included various patients after multi-trauma surgeries (neurosurgery, thoracostomies; exploratory laparotomy for the liver and/or spleen damage; orthopedics, osteosynthesis; severe facial fractures and vascular damage related to open fractures). Two neurosurgery patients in the placebo group developed SSI, whilst only one neurosurgery patient in the probiotic group developed SSI. Due to such a low incidence of SSI, statistical analysis was not possible. Among all included surgical patients, *Staphylococcus aureus* and *Acinetobacter baumannii* were the most common pathogens. A significantly lower incidence of SSI after MT surgeries in the probiotic group compared to the placebo group (*p =* 0.022) suggests that prophylactic administration of probiotics in MT patients exerts a positive effect on the incidence of SSI.

* new nomenclature of *lactobacilli* according to [49]; bid: twice daily; EEN: early enteral nutrition; MT: multi-trauma; RCT: randomized controlled trial; SSI: surgical site infections.

**Table 3 microorganisms-12-01361-t003:** Findings of the four studies investigating the effect of probiotics on patients after traumatic brain injury, in descending chronological order.

Reference	Aim and Study Type	Participants Who Completed Study	Intervention		Main Findings
			Probiotics	Dosage/Duration	
Zhang et al., 2021 [41], China	Double-blind RCT to assess the effect of EEN with probiotics on patients after TBI.	136 patients after TBI. 68 in probiotic group. 68 in control group.	*Bifidobacterium longum* **, *Lactobacillus acidophilus* **, *Enterococcus faecalis* **. (Bifid Triple Viable Enteric-Coated Capsules, Jincheng Haisi Pharmaceutical, Jincheng, China).	Dosage: two capsules bid. Cfu: 3 × 10^6^ cfu/g. Total cfu per day: ND. Duration: 14 days.	Although both groups showed notable changes in TBI patients after EEN, more significant changes related to the reduction in inflammatory response and enhanced immune function were observed in the probiotic group compared to the placebo group (*p* < 0.05). The incidence rate of complications was also evidently lower in the probiotic group compared to the placebo group (*p* < 0.05).
Wan et al., 2020 [42], China	Double-blind RCT to assess the effect of EN with probiotics on patients after TBI.	76 patients with severe TBI. 38 in probiotic group. 38 in control group.	*Bifidobacterium longum* **, *Lactobacillus bulgaricus* **, *Enterococcus faecalis* ** (Xinyi Pharmaceutical Factory, Shanghai, China).	Dosage: six tablets bid. Total cfu per day: 1.2 × 10^8^ cfu. Duration: 15 days.	Interleukin (IL)-6, IL-10, tumor necrosis factor (TNF)-a, and CRP at 7 and 15 days decreased significantly more in the combined treatment group. Thus, probiotics together with EN improve the recovery of patients with severe TBI. Hospitalization duration and pulmonary infection rates were also significantly reduced in the combined compared with the EN alone group. GCS scores at 15 days were significantly lower in the combined treatment group compared with the EN group.
Tan et al., 2011 [43], China	Double-blind RCT to assess the effect of probiotics after severe TBI.	52 patients with severe craniocerebral trauma. 26 in probiotic group. 26 in control group.	*Bifidobacterium longum* **, *Lactobacillus bulgaricus* **, *Streptococcus thermophilus* **. (Golden bifid Shuangqi Pharmaceutical, Inner Mongolia, China).	Dosage: 7 sachets tid. Total cfu per day: 1.0 × 10^9^ cfu. Duration: 21 days.	The probiotic group exhibited a significantly higher increase in serum IL-12p70 and IFN-γ levels, coupled with a dramatic decrease in IL-4 and IL-10 concentrations, as compared to the control group. Patients in the probiotic group experienced a decreased incidence of nosocomial infections towards the end of the study. Shorter ICU stays were also observed among patients treated with probiotic therapy.
Falcao de Arruda et al., 2004 [44], Brazil	RCT to assess the effect of EEN with glutamine and probiotics on patients with brain injury.	20 patients with brain injury. 10 in probiotic group. 10 in control group.	*Lactobacillus johnsonii* La 1. (LC1^®^; Nestle, Sao Paulo, Brazil).	Dosage: 240 mL fermented milk with La 1. Total cfu per day: 10^9^ cfu. Duration: 5 to 14 days.	Significantly lower levels of infection rate (*p* = 0.03), number of infections per patient (*p* < 0.01), number of days in the intensive care unit (*p* < 0.01) and days of mechanical ventilation (*p* = 0.04) of brain injury patients were observed in the group that received probiotic and glutamine and EEN compared to the control group that received enteral nutrition only.

** no strain information in study; bid: twice daily; EEN: early enteral nutrition; EN: enteral nutrition; GCS score: Glasgow Coma Scale score; ND: no data; RCT: randomized controlled trial; TBI: traumatic brain injury; tid: three times daily; qd: once daily.

**Table 4 microorganisms-12-01361-t004:** Findings of four studies investigating the effect of probiotics on patients after spinal surgery or back pain, in descending chronological order.

Reference	Aim and Study Type	Participants who Completed Study	Intervention		Main Findings
			Probiotics	Dosage/Duration	
Kaku et al., 2020 [45], Japan	Double-blind RCT to assess the effect of probiotics in patients administered SAP before spinal surgery.	33 patients after spinal surgery. 16 in probiotic group. 17 in control group.	*Enterococcus faecium* 129 BIO 3B-R. (Biofermin Pharamceutical Ltd., Kobe, Hyogo, Japan)	Dosage: 1 g tid. Total cfu per day: ND. Duration: 10 days.	After evaluation of gut microbiome, no significant differences were found in alpha and beta diversity. However, *Streptococcus gallolyticus* and *Roseburia* were significantly decreased in the probiotic group compared with the control group. Considering the pathogenicity of *Streptococcus gallolyticus*, SAP had a negative influence on patients, and probiotics could prevent possible adverse effects after surgery.
Taye et al., 2020 [46], Australia.	Case study to assess the effect of probiotics for osteoarthritis, including lower back and ankle pain.	1 patient with osteoarthritis, including lower back pain.	*Lacticaseibacillus rhamnosus* GG *Saccharomyces cerevisiae* var. *boulardii* *Bifidobacterium animalis* subsp. *lactis* BB-12. (Metagenics Ultra Flora Intensive Care, East Lismore, Australia).	Dosage: 2 capsules bid. Total cfu per day: 10^11^ cfu. Duration: 3 blocks of 10 weeks with 2 weeks follow-up period.	The probiotic intervention was associated with lower pain scores and was the preferred intervention chosen by the participant. The mean pain score on the VAS scale was 4.9 ± 2.2 in the control group compared to 4.0 ± 1.7 in the probiotic group (*p* = 0.04). Although the reduction in pain scores associated with probiotic intervention was small, it was clinically significant for this patient.
Jensen et al., 2019 [47], Denmark	Double-blind RCT to assess the effect of probiotics on chronic low back pain.	85 patients with chronic low back pain. 42 in probiotic group. 43 in control group.	*Lacticaseibacillus rhamnosus* GG. (Dicoflor^®^, Rome, Italy)	Dosage: 1 capsule bid. Total cfu per day: 1.2×10^10^ cfu. Duration: 100 days.	A small, though hardly clinically relevant, effect on back pain using the VAS scale was seen after supplementation with probiotics compared to the control group. Back pain was statistically significantly reduced in the probiotic group (from 6.0 ± 1.81 to 4.7 ±2.57) compared to the control group (from 5.6 ± 1.63 to 5.3 ±2.00) after a one-year follow-up.
Shukla et al., 2016 [48], India	Double-blind RCT to assess the effect of probiotics on the immune and clinical parameters of children having JIA-ERA, including back pain.	40 children with JIA-ERA. 21 in probiotic group. 19 in control group.	*Streptococcus thermophilus* **, *Bifidobacterium breve* **, *Bifidobacterium longum* **, *Bifidobacterium infantis* **, *Lactobacillus acidophilus* **, *Lactiplantibacillus plantarum* **, *Lacticaseibacillus paracasei* **, *Lactobacillus delbrueckii* *. (VSL#3, Sun Pharmaceuticals, Mumbai, India)	Dosage: 1 capsule bid. Total cfu per day: 2.25 × 10^11^ cfu. Duration: 12 weeks.	No significant difference was observed in the improvement in the six-point composite disease activity index (mJSpADA) in the probiotics group compared to the control group. A significant decrease in the inflammatory cytokine Il-6 was observed in the probiotic group. Th2 cell frequency and serum IL-10 levels showed an increase in the control group, but the probiotic use did not show a significant change in immune parameters when compared to the control group.

* new nomenclature of *lactobacilli* according to [49]; ** no strain information in study; bid: twice daily; JIA-ERA: juvenile idiopathic arthritis—enthesitis-related arthritis, ND: no data; RCT: randomized controlled trial; SAP: Surgical antibiotic prophylaxis; tid: three times daily; VAS: Visual Analogue Scale.

## Data Availability

No new data were created.

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
