# Peer review of "Overview of the Efficacy of Using Probiotics for Neurosurgical and Potential Neurosurgical Patients"

_microorganisms, 2024, doi:10.3390/microorganisms12071361_

Round 1

Reviewer 1 Report

Comments and Suggestions for Authors

In the paper entitled "Overview of the efficacy of using probiotics for neurosurgical and potential neurosurgical patients", the authors made a comprehensive literature review and identified 10 relevant publications focusing on probiotic interventions for various neurosurgical conditions across multiple countries. They concluded that although probiotics offer a promising adjuvant therapy for neurosurgical patients, traumatic brain injuries, and post-spinal surgery, further well-designed randomized controlled trials are essential to elucidate the intricate relationship between microbiome-modulating interventions and the central nervous system via the gut microbiota-brain axis.

Although the paper is interesting one major problem is the high plagiarism percentage mainly on the introduction were entire paragraphs were copied form other sources.

Comments on the Quality of English Language

Moderate editing of English language required

Author Response

Dear reviewer. Thank you for your report. We have done our best to address all your comments. Please find the answers in the attachment. The revised manuscript has been resubmitted and all the corrections are noted in yellow.

Reviewer 2 Report

Comments and Suggestions for Authors

In this review, Fijan et al. explored the ‘gut microbiota-brain axis’ and its role in neurological health, particularly for neurosurgical patients, highlighting the potential of probiotics to improve outcomes like gastrointestinal mobility, infection rates, and inflammatory responses.

This review presents a meaningful topic and the data was well organized, but there are some minor issues. Here are some comments on this study:

1.        Please define the abbreviation when it first appears, such as line 58 “central nervous system”.

2.        Line 70 “primary and secondary bile acid (BAs) pathways”, primary BAs refer to synthesized BAs by the host, not metabolic pathways of the gut microbiota.

3.        Line 103 “(Ratna et al., 2023)” a different type of citation.

4.        It would be good to mention the meaning or significance of this review at the end of the introduction.

5.        Section 2 “Materials and Methods”, it would be useful to have a diagram to describe the process of this review study.

6.        Lines 123-125 “The results of the 123 studies in table 1 therefore confirm the positive influence of probiotic supplementation on 124 immunological response”, I don't see results on immunization in Table 1.

Author Response

(The authors gave the same response as above.)

Reviewer 3 Report

Comments and Suggestions for Authors

Peer Review Report

Peer review report 1 on ‘‘Overview of the efficacy of using probiotics for neurosurgical and potential neurosurgical patients’’.

1. Original Submission

     1.1. Recommendation: Minor Revision

2. Comments to authors

Overview and general recommendation:

Overall, the study is well designed, well performed and clearly presented. Although the flaws within the manuscript, I suggest its publication in case of minor revision. Some indications for minor revisions are given below.

* You did not put any figure in the text. Try, at least to add a graphical abstract in order to sum up the work!

* Try to briefly mention the aim of the study within the abstract!

* Line 65: you have mentioned the link between probiotics and different organs in the context of various axis, it would be a plus if you add the following axis: 'Probiotics-Sex hormones axis', notably in the line 67 you have mentioned the endocrine system. 

* Line 103: You should write all the references with the same style, as recommended by the journal [x].

* Page 7 of 15 (line 1): ' The findings of the 10 studies noted in tables 1 and 2 were very diverse after...', you miss to mention the table 3!

* Page 8 of 15 (line 102): ' Three studies our review assessed...', it should be corrected as following: ' Three studies of our review assessed...'.

* In the 'Discussion' section, try to avoid repeating what was written in tables in terms of redescribing all the investigations. Try to deeply discuss and compare those conducted studies with other connected investigations and extrapolate on probiotic use in other fields (in treating and/or preventing diseases).

* The discussion section needs to be partially restructed, in terms of retreatment and rediscussion of the ideas and scientific literature.

* Try to develop a section entitled 'Conclusion and Perspectives' in which you may open the gate to other axis in human body and underscore the role of probiotics in boosting health via their firm connection to these different axis.

* Check all the text in order to put commas and punctuation in right places to give the meaning to the sentences and ideas.

* Check all the text for italic mode.

Comments on the Quality of English Language

The English style is fine, but it needs to be refined all through the text.

Author Response

(The authors gave the same response as above.)

Reviewer 4 Report

Comments and Suggestions for Authors

Dear authors,

It is an interesting study, which has difficulties in designing, due to only a few studies on the subject and, each one with different parameters and different probiotics used.

my comments:

1. in my opinion, the introduction, although comprehensive, is quite lengthy

2. regarding the 2nd paper [Table 1] by Tzikos et al:  OK, there are only 6 and 13 patients being subjected to craniotomy, in probiotic and control groups; however all 103 patients have a severe brain trauma [being a prerequesite to enter the study - multi trauma patients having severe brain trauma plus at least one more trauma] the severity of which led to the immediate intubation. Thus, I consider that this information should be mentioned a appropriately commented, since the topic in not only neurosurgery but severe traumatic brain injury

Author Response

(The authors gave the same response as above.)
